# Unsupervised Embedding Learning for Large-Scale Heterogeneous Networks Based on Metapath Graph Sampling

**DOI:** 10.3390/e25020297

**Published:** 2023-02-04

**Authors:** Hongwei Zhong, Mingyang Wang, Xinyue Zhang

**Affiliations:** College of Information and Computer Engineering, Northeast Forestry University, Harbin 150040, China

**Keywords:** large-scale heterogeneous network, network embedding learning, unsupervised learning, metapath, mutual information

## Abstract

How to learn the embedding vectors of nodes in unsupervised large-scale heterogeneous networks is a key problem in heterogeneous network embedding research. This paper proposes an unsupervised embedding learning model, named LHGI (Large-scale Heterogeneous Graph Infomax). LHGI adopts the subgraph sampling technology under the guidance of metapaths, which can compress the network and retain the semantic information in the network as much as possible. At the same time, LHGI adopts the idea of contrastive learning, and takes the mutual information between normal/negative node vectors and the global graph vector as the objective function to guide the learning process. By maximizing the mutual information, LHGI solves the problem of how to train the network without supervised information. The experimental results show that, compared with the baseline models, the LHGI model shows a better feature extraction capability both in medium-scale unsupervised heterogeneous networks and in large-scale unsupervised heterogeneous networks. The node vectors generated by the LHGI model achieve better performance in the downstream mining tasks.

## 1. Introduction

Embedding learning based on heterogeneous networks can integrate multi-dimensional heterogeneous information in the network and generate node vectors with rich semantics. The supervised embedding learning models can optimize the parameters in the model with the help of supervision information, and extract the features in the network to encode the nodes [1,2,3]. However, in practice, because it is difficult to label large-scale network data, most heterogeneous networks do not contain supervision information. This makes how to optimize the learning model without supervision information become the key problem in heterogeneous network learning tasks. 

Various unsupervised embedding learning models have been proposed to encode nodes in heterogeneous networks, such as the models based on random walk [4,5,6], Generative Adversarial Network (GAN) [7,8], autoencoder model [9,10] and graphical neural network (GNN) [11]. However, when faced with a heterogeneous network composed of millions of nodes, these models have difficulty overcoming the problem of high space-time complexity caused by the large network scale, which limits the application of these models on large-scale heterogeneous networks. Researchers proposed a graph sampling technique [12,13] to solve this problem. In the original large-scale heterogeneous network, a batch graph sampling technology is used to sample the network substructures, and learn node embeddings by training the substructures. Although graph sampling technology can help compress the network, it only extracts some substructures from the original network, which leads to the loss of network information. The subsequent embedding process can only learn the vectors of nodes on the features in the substructures, which affects the application of node embeddings in downstream mining tasks.

In this paper, we propose a large-scale heterogeneous network embedding learning model LHGI (Large-scale Heterogeneous Graph Infomax) based on metapath graph sampling technology. LHGI maps the initial large-scale heterogeneous network into a series of homogeneous graphs under the guidance of metapaths. Each homogeneous graph reflects the neighborhood relationship between nodes on a certain metapath. LHGI generates the sampling graphs by executing an iterative sampling strategy on the homogeneous graphs. Because the information extracted from different metapaths is integrated in the sampling process, the sampling graph obtained can integrate the low-order and high-order semantic relationships between nodes in the initial heterogeneous network, thus avoiding the loss of information in the network as much as possible. Then a contrastive leaning process is conducted to generate the embedding vectors of nodes in the network. The main contributions of LHGI model are as follows:(1)LHGI is an embedding learning model designed for large-scale heterogeneous networks. LHGI adopts the graph sampling strategy under the guidance of metapaths, so that the model can compress the network while preserving the low-order and high-order neighbor relationships between nodes as much as possible. This provides a guarantee for the model to generate high-quality node embedding vectors.(2)LHGI is an unsupervised embedding learning model. LHGI introduces the idea of contrastive learning, which enables LHGI to optimize parameters in the model, so that the embedding vectors of nodes can be learned without any supervision information.

## 2. Related Work

Researchers initially introduced random walk technology to solve the unsupervised problem in network embedding. By executing random walking in the network, a series of node sequences are generated, which are input into the skip-gram model to generate the embeddings of nodes in the sequence. Following this strategy, Deepwalk [14], Node2vec [15], Struct2vec [16] and other models were proposed. Subsequently, the unsupervised model based on the autoencoder was introduced to encode nodes. The SDNE [17] model optimizes the first-order and second-order similarity between nodes through the self-encoder structure to learn the vectors of nodes. The VGAE [18] model draws on the idea of graph encoding and decoding, and trains the embedding process of nodes by minimizing the error between the original graph and the reconstructed graph. The DIM [19] model maps an image as a sequence reflecting image structural features, and encodes the image by maximizing mutual information between feature mapping and global feature vectors. DGI [20] uses a strategy of maximizing the mutual information between node vectors and global map vectors to learn node embeddings.

However, the above models are proposed for unsupervised embedding learning tasks in homogeneous networks. Heterogeneous networks contain a variety of relationships between nodes of different types. This makes the information transmission between heterogeneous nodes more complex, and also makes it difficult to directly migrate the learning models suitable for homogeneous networks to heterogeneous networks. Many embedding learning models were proposed for heterogeneous networks, where metapaths have been leveraged by most of the existing works on heterogeneous network modeling [21].

Researchers designed metapaths based on node types and their relationships to guide the process of random walk. For example, the Metapath2vec model [22], HERec model [23] and Hin2vec model [24] are all models under the guidance of metapaths. However, the strategy of guiding random walk based on metapaths becomes time-consuming with the growth of the number of nodes and edges, and the node embedding results are vulnerable to the walk strategy. After that, researchers introduced the Generative Adversarial Network (GAN) to encode nodes under the unsupervised learning. GAN can approach the distribution of the original data under the idea of a game. In the HeGAN model [25] and GCR-GAN model [26], the node embeddings are learned without supervision information by virtue of the game process between the generator and discriminator. However, in the face of large-scale network data, the GAN model is prone to slow training speed and difficulty in converging. Besides GAN, graph convolution (GCN) technology is also used to learn the embeddings of nodes. The HDGI model [27] uses GCN technology to aggregate the information of neighbor nodes, so as to generate the embedding of target nodes. However, with the increase of the network size, the matrix operation in the graph convolution process becomes time- and memory-consuming.

In order to deal with the problem of high space and time complexity caused by the growth of network size, researchers proposed graph sampling technology. For homogeneous networks, the GraphSAGE model [28] was proposed, which samples a certain number of neighbor nodes for each target node. For heterogeneous networks, the HGT model [29] was proposed, which uses the HG Sampling algorithm of small batches to reserve the same number of heterogeneous neighbor nodes and edges for target nodes.

It can be seen that researchers have made in-depth explorations on the node embedding task in unsupervised heterogeneous networks. However, these explorations are only applicable to medium-scale heterogeneous networks, and it is still difficult to directly migrate them to large-scale heterogeneous networks. Although graph sampling technology can compress the size of the network, it can only sample the local substructure of the target node. In other words, this technology can only obtain the low-level neighbor relationship of the target node, which leads to the loss of higher-order semantic information in the network, thus ultimately limiting the performance of node embedding.

In this paper, we propose an unsupervised embedding learning model, LHGI, for large-scale heterogeneous networks. LHGI uses a graph sampling technology guided by metapaths to compress the size of network. Compared with the existing graph sampling technologies, the metapath guiding graph sampling technology can capture low- and high-order semantic information in large-scale heterogeneous network. At the same time, LHGI uses the idea of contrastive learning to optimize model parameters, which enables the model to learn the embedding of nodes without supervision information.

## 3. Overall Architecture of the LHGI Model

### 3.1. Definitions

Definition 1: Heterogeneous network

A heterogeneous network is a graph composed of different types of nodes and different types of edges between nodes. Formally, a heterogeneous network can be defined as $G=\left\{V,E,A,R\right\}$. Where V is the set of nodes, E is the set of edges, and A and R are the set of node types and edge types, respectively. The mapping function of node type is: ϕ: V→A. The mapping function of edge type is: φ:E→R.

Definition 2: Metapath

In a heterogeneous network, a metapath is a path composed of a sequence of relationships between heterogeneous nodes. Formally, a metapath Φ can be defined as $v_{1}\overset{R_{1}}{\to }v_{2}\;\overset{R_{2}}{\to }\cdots \overset{R_{n-1}}{\to }v_{n}$, where $R=R_{1}\circ R_{2}\circ \cdots \circ \;R_{n-1}$ defines the sequence relationship between nodes $v_{1}$ and $v_{n}$. A metapath “*A-P-A*” can be labeled as $\Phi_{APA}$.

### 3.2. Framework of LHGI Model

Figure 1 shows the overall architecture of the LHGI model. LHGI consists of four modules: Meta Path Generation Module, Meta Path guided Graph Sampling Module, Sampling Graph Aggregation Encoding Module and Discriminator Module. In the Meta Path Generation Module, a series of metapaths is automatically generated by performing progressive multiplication between initial adjacency matrices of a heterogeneous network. Four filtering principles are proposed to screen out metapaths with high semantic information. Each metapath can be mapped into a homogeneous graph. In the Meta Path guided Graph Sampling module, LHGI samples nodes and edges in the homogeneous graph in batches to obtain the sampling graph. On the sampling graphs, a GCN encoder and attention mechanism are used to learn the embedding vectors of nodes, and on this basis, the embedding vector of the global network graph is generated. At the same time, the Negative Sample Generator in the sampling module is used to generate negative sample nodes, so that LHGI can optimize the parameters in the model through contrastive learning between normal nodes and negative sample nodes. In the Discriminator module, LHGI adjusts the parameters in the model to learn the node vectors by maximizing the mutual information between the node vectors and the global network vector.

#### 3.2.1. Meta Path Generation Module

There is a unique one-to-one correspondence between the adjacency matrix and the topology of the network. Through the progressive multiplication of adjacency matrices, a series of metapaths is automatically generated. In order to exclude redundant and low semantic metapaths, four screening principles are designed in this paper.

Principle 1: In one metapath, up to two nodes in a certain type can exist.

$$num\_nodes\left(\Phi \right)\left(V\in A_{i}\right)\leq 2,$$
where Φ is a metapath, $A_{i}$ is a certain node type, and V represents the nodes in $A_{i}$.

Principle 2: In one metapath, up to three different types of nodes can exist.

$$num\_types\left(\Phi \right)\left(A\right)\leq 3,$$
where Φ is a metapath, and A is the set of node types.

Principle 3: Metapaths containing a subpath with 1:1 relationship between its head and tail node will be excluded.

$$ER\langle \varphi \rangle \neq \langle 1:1\rangle ,$$
where φ is a sub-path in a metapath Φ, and $ER\langle \varphi \rangle$ indicates the relationship between the head and tail nodes in φ.

Principle 4: Only metapaths with the same type of head and tail nodes are retained.

$$symmetry\left(\Phi \right)=True,$$
where Φ is a metapath. $symmetry\left(\Phi \right)$ is used to determine whether the head and tail node in Φ are of the same type. 

Principle 4 is set to preserve the metapath adjacency relationship between homogeneous nodes. If the head and tail nodes belong to different node types, the adjacency semantic relationship between them is complex and difficult to explain. Therefore, only the metapath adjacency relationship between homogeneous nodes is retained.

Based on the four principles, the metapaths that meet the requirements are screened out. Then the relationship between the head and tail nodes in the metapath is extracted and mapped into a homogeneous graph. A metapath maps to one homogeneous graph. The homogeneous graph can well preserve the information transmission between homogeneous nodes on a specific metapath, and can preserve the low-order or high-order semantic information in the heterogeneous network according to the different lengths of metapaths.

#### 3.2.2. Meta Path Guided Graph Sampling Module

In order to deal with the high space-time complexity in training a large-scale network, this paper designs a graph sampling module in LHGI. Unlike the existing sampling techniques, which directly sample the local substructure in network, the graph sampling process proposed in this paper is based on the homogeneous graphs mapped by the metapaths. In the homogenous graph, the adjacency relationship between nodes actually reflects the information transmission between the head and tail nodes in the metapath. The sampling process based on the homogenous graph can retain the semantic information of nodes on the metapath, rather than the first-order adjacency relationship in the initial heterogeneous network. According to the different lengths of metapaths, graph sampling based on metapaths can retain low and high order semantic information between nodes in the network. Therefore, compared with the strategy of directly sampling the neighborhood substructure of the target node, the graph sampling strategy under the guidance of metapaths proposed in this paper is more helpful to retain the higher-order semantic relations in the network, so as to avoid the loss of information caused by the sampling process as much as possible.

For all the nodes in the homogeneous graph, the resampling method is used to sample a fixed number of neighbor nodes for each node. For target node v, define the set $S_{i}=\{S_{i-1}\cup u\;|\;u\in N\left(S_{i-1}\right)\}$ to represent the node set obtained by uniformly resampling the first-order neighborhood of the nodes in the set $S_{i-1}$, where $N\left(v\right)$ represents the set of all first-order neighbor nodes of node v, and i represents the number of iterative samplings $i\in \left\{1,\ldots ,k\right\}$. Specially, $S_{0}=\left\{v\right\}$ denotes node v itself, and $S_{1}=\{v\cup u\;|\;u\in N\left(v\right)\}$ represents the node set after sampling the first-order neighbor of v once. $S_{2}=\{S_{1}\cup u\;|\;u\in N\left(S_{1}\right)\}$ is the node set after sampling the first-order neighbor of the nodes in $S_{1}$ once, and so on. The sampling set of neighbor nodes from order 1 to order k of the target node can be obtained through k times of sampling. The total space complexity and time complexity of the algorithm for k times sampling is both $O\left(\prod_{i=1}^{k}S_{i}\right)$. The value of k is consistent with the number of encoder layers by default. For target node x, we obtain its neighborhood $\left\{S_{k}^{\Phi_{j}}\;|\;j\in Set\left(\Phi \right)\right\}$ under n metapaths according to the above sampling process, where $Set\left(\Phi \right)$ represents the collection of metapaths obtained in the Meta Path Generation Module, and $S_{k}^{\Phi_{j}}$ is the subgraph obtained after k times of sampling under the metapath $\Phi_{j}$ for target node x.

Figure 2 shows the schematic diagram of the graph sampling process on the three metapaths of target node x, where $Set\left(\Phi \right)=\left\{\Phi_{X},\Phi_{Y},\Phi_{Z}\right\}$ represents three metapaths, and the target node x appears in these metapaths as the head or tail node. On the homogenous graphs mapped by the three metapaths, the sampling process is executed respectively. The uniform distribution sampling method is used to obtain the neighborhood set of a single node by repeatable sampling from its neighbors. The size of the neighborhood set is a hyperparameter in the LHGI model, which is 20 by default. After the first sampling, we get the first-order neighbors of x in the homogeneous graph: $\left\{S_{1}^{\Phi_{X}},S_{1}^{\Phi_{Y}},S_{1}^{\Phi_{Z}}\right\}$. These neighbors are not the real first-order neighbors of the target node x in the initial heterogeneous network, but the neighbor nodes that have the head-to-tail correspondence with x in the metapath. Next, taking the neighbor nodes obtained in the first sampling as seed nodes, the neighbor nodes of these seed nodes under the metapath are sampled to generate the second sampling set: $\left\{S_{2}^{\Phi_{X}},S_{2}^{\Phi_{Y}},S_{2}^{\Phi_{Z}}\right\}$. The second sampling will obtain the second-order neighbors of target node x in the homogeneous graph. Similarly, these neighbors are not the initial second-order neighbors of x, but the second-order neighbors in the sense of metapath. After k times of sampling, we can get: $\left\{S_{k}^{\Phi_{X}},S_{k}^{\Phi_{Y}},S_{k}^{\Phi_{Z}}\right\}$. Then the three sampling sets of $S_{k}^{\Phi_{X}},S_{k}^{\Phi_{Y}}$ and $S_{k}^{\Phi_{Z}}$ are input into the Sampling Graph Aggregation Encoding Module in parallel, based on which to learn the embedding vectors of node x.

#### 3.2.3. Sampling Graph Aggregation Encoding Module

GCN can capture the local topology characteristics of nodes in the network without prior information, so GCN is used to learn the embedding vectors of nodes in this paper. For each sampling graph, GCN encodes the target node by aggregating the information of its “neighbors”. The neighbors here are no longer the first-order neighbors of the target node in the initial heterogeneous network, but the higher-order neighbors determined by the head and tail relationship in the metapaths. And because the sampling graph is obtained by sampling the metapath many times, the sampling process can approach the deeper semantic information contained in the initial heterogeneous network as much as possible. The formula of GCN is as follows:(1)$$H^{\Phi_{i}}=\left(D^{\Phi_{i}-\frac{1}{2}}A^{\Phi_{i}}D^{\Phi_{i}-\frac{1}{2}}\right)XW^{\Phi_{i}},$$
where $H^{\Phi_{i}}$ refers to the embedding vector of target node under the graph sampled by metapath $\Phi_{i}$, X represents the initial characteristic matrix of the node, $D^{\Phi_{i}}$ is the degree matrix under metapath $\Phi_{i}$, $A^{\Phi_{i}}$ is the adjacency matrix under $\Phi_{i}$, and $W^{\Phi_{i}}$ is the parameter matrix. 

A given node may appear in different metapaths, which means that one node will obtain different embedding vectors. Here, the semantic attention mechanism is used to measure the weight of semantic information provided by different metapaths. According to the weights of the metapaths, the node vectors generated by different metapaths are aggregated to obtain the final embedding vector of the node. The following formulas show the aggregation process.
(2)$$H=\overset{P}{\underset{i=1}{\sum }}Att_{\Phi_{i}}\cdot H^{\Phi_{i}}$$
(3)$$Att_{\Phi_{i}}=Softmax\left(W_{\Phi_{i}}\right)=\frac{\exp\left(W_{\Phi_{i}}\right)}{\sum_{j=1}^{P}\exp\left(W_{\Phi_{i}}\right)}$$
(4)$$U_{\Phi_{i}}=\mathrm{Tanh}\left(H^{\Phi_{i}}W+B\right)$$
(5)$$W_{\Phi_{i}}=U_{\Phi_{i}}\cdot Q^{T},$$
where H is the final vector matrix learned. H is obtained by weighted aggregation of the node embedding matrix $H^{\Phi_{i}}\;$under the weight matrix $Att_{\Phi_{i}}$ of the metapaths. $W_{\Phi_{i}}$ represents the weight matrix of the metapath under the self-attention mechanism. The Softmax function is used to normalize $W_{\Phi_{i}}$ to obtain the weight matrix $Att_{\Phi_{i}}\;$of the metapaths. $W_{\Phi_{i}}$ is obtained by multiplying the key vector matrix, $U_{\Phi_{i}},\;$and the query vector matrix, $Q^{T}$. The vector matrix $U_{\Phi_{i}}\;$is obtained by $H^{\Phi_{i}}$ through a layer of MLP mapping, which uses Tanh as the activation function. W, B and $Q^{T}$ are the training parameters.

At the same time, a Negative Sample Generator is used to generate negative sample nodes by randomly disrupting the rows in the initial node characteristic matrix X, which will disrupt the order of node indexes and destroy the connection between nodes. Then, the same embedding learning process as for the normal sample nodes is performed to learn the embedding vectors of the negative sample nodes. Because the initial characteristics of the negative sample nodes and their adjacent relationships with other nodes are randomly disturbed, the negative sample node vector is not learned on the real network space features. This makes the vector of the negative sample node significantly different from that of the normal node, so the LHGI model can solve the problem of unsupervised training through the idea of comparative learning.

Then the vector of global graph is generated by aggregating the local graph vectors calculated in each batch. The local graph vector is obtained by averaging the vectors of nodes obtained from each batch of sampling. Figure 3 shows the schematic diagram of generating the global map vector. In the Discriminator Module of the LHGI model, the global map vector is used to calculate the mutual information with the node vectors, so that the LHGI model can be trained by maximizing the mutual information.

#### 3.2.4. Discriminator Module

Mutual information is often used to measure the amount of information shared between two random variables. Let the mutual information between two random variables X and Y be $I\left(X,Y\right)$, which is usually defined as:
(6)$$\begin{array}{c} I\left(X,Y\right)=\sum_{X,Y}P\left(X,Y\right)log\frac{P\left(X|Y\right)}{P\left(X\right)}\;\\ \;\;\;\;\;\;\;\;\;\;=D_{KL}(P\left(X,Y\right)||P\left(X\right)P\left(Y\right)). \end{array}$$

Greater mutual information means that the variable X will provide more effective information to the variable Y. In practice, because it is hard to accurately calculate mutual information, researchers have proposed some estimation strategies. In the DIM [19] model, researchers proposed the MINE (Mutual Information Neural Estimation) method, which gives the lower bound of mutual information based on KL divergence:(7)$$I\left(X,Y\right)=D_{KL}\left(J||M\right)\geq {\hat{I}}_{\omega }\left(X,Y\right)=E_{J}\left[T_{\omega }\left(x,y\right)\right]-logE_{M}\left[e^{T_{\omega }\left(x,y\right)}\right],$$
where $J=P\left(X,Y\right)$, $M=P\left(X\right)\cdot P\left(Y\right)$, and T is a deep neural network-based discriminator parametrized by ω. Specifically, for the LHGI model, NCE (Noise Contrastive Estimation) loss is used to approximately and equivalently estimate the boundary of mutual information:(8)$$L=\frac{1}{N+M}\left(\overset{N}{\underset{i=1}{\sum }}E_{\left(X,\vec{s}\right)}\left[\log D\left({\vec{h}}_{i},\vec{s}\right)\right]+\overset{M}{\underset{j=1}{\sum }}E_{\left(\tilde{X,}\vec{s}\right)}\left[\log\left(1-D\left({\vec{\tilde{h}}}_{j},\vec{s}\right)\right)\right]\right),$$
where $\vec{s}$ represents the vector of the global graph, ${\vec{h}}_{i}$ is the vector of the normal node $x_{i}$, and ${\vec{\tilde{h}}}_{j}$ is the vector of the negative sample node. By learning the difference between the original node sample distribution and the negative sample distribution, the structural characteristics of the original nodes in the heterogeneous network can be learned in the process of maximizing the mutual information.

The discriminator D uses the bilinear model to calculate the joint probability between the global map vector $\vec{s}\;$and the target node vector ${\vec{h}}_{i}$: (9)$$D\left(\vec{h_{i}},\vec{s}\right)=\mathrm{sigmoid}\left(\vec{s}\cdot W\cdot {\vec{h}}_{i}^{T}\right).$$

#### 3.2.5. Complexity Analysis

The spatial complexity of the LHGI model is mainly determined by the Meta Path Generation Module, Meta Path guided Graph Sampling Module and Sampling Graph Aggregation Encoding Module. The number of metapaths obtained in the Meta Path Generation Module determines the number of GCN encoders required by LHGI. In the Meta Path guided Graph Sampling Module, the number of target nodes processed by each batch, the number of neighbor nodes sampled by each batch and the number of sampling batches are the main factors affecting the input scale of LHGI. In the Sampling Graph Aggregation Encoding Module, LHGI uses a two-layer GCN encoder to avoid the over-smoothing problem in the learning process [30]. Suppose the number of metapaths is a, the number of target nodes processed in each batch is b, the number of neighbors sampled in each batch is c, and the number of sampling iterations is d (which is equal to the number of layers of GCN encoder by default), then the spatial complexity of LHGI model is$\;O\left(a{\left(bc\right)}^{d}\right)$. 

The time complexity of the LHGI model is determined by the running time of each batch and the number of batches. The number of batches is related to the number of target nodes processed under each batch. The more target nodes processed in each batch, the less batches required, and vice versa. Therefore, the time complexity of the LHGI model can be expressed as $O\left(b^{-1}t\right)$, where b is the number of target nodes processed by each batch, and t is the running time for each batch. 

Figure 4 shows the relationship between batch size and batch (epoch) runtime. The X coordinate represents the batch size, that is, the number of target nodes processed under each batch. The Y coordinate represents the running time of each batch in Figure 4a) and each epoch in Figure 4b), respectively. It can be seen from Figure 4a) that, with the increase of the number of target nodes processed by each batch, the running time of each batch will increase. However, as the number of target nodes processed in each batch increases, the number of batches will decrease, which leads to the reduction of running time in each epoch, as shown in Figure 4b). 

For the two large-scale heterogeneous networks given in Section 4.2, the running time of AMiner in a batch is slightly shorter than that of ogbn-mag under the same batch size. This is because fewer metapaths are generated in the AMiner network. However, under the same batch size, the running time of AMiner in an epoch is higher than that of ogbn-mag. The reason is that there are more target nodes to be processed in AMiner than that in ogbn-mag. Accordingly, the number of batches needed to train in AMiner is more than that in ogbn-mag under the same batch size. Therefore, it takes longer time to train AMiner in an epoch. 

Taking the AMiner network as an example, when the batch size is 1024, the running time of an epoch is 109.5 s. When the batch size is 131,072, the running time of an epoch is 42.16 s. It can be seen that for large-scale heterogeneous networks, the LHGI model can easily train the network under different batch sizes. In practical applications, the appropriate batch size can be selected according to the hardware environment, which enables LHGI model more universal.

## 4. Experimental Results & Discussions

### 4.1. Baseline Models

To verify the performance of the LHGI model, some unsupervised heterogeneous network embedding learning models are taken as the baseline models to compare with LHGI. These baseline models include the models based on random walk architecture, models based on GAN architecture, and models based on graph sampling architecture. 

Metapath2vec(M2V) [22]: M2V is a model based on random walk, which takes Word2vec model as its backbone architecture.Hin2vec [24]: Hin2Vec is a model based on metapath random walk, which uses a logical binary classifier to predict whether there is a specific relationship between two given nodes to guide the learning process.HeGAN [25]: HeGAN is a model based on contrastive learning, which uses a generator to generate pseudo nodes, and uses a discriminator to identify the authenticity of nodes.HDGI [27]: HDGI uses a metapath to capture the semantic structure in a heterogeneous network, and uses graph convolution modules and semantic level attention mechanisms to learn the embedding of nodes.HGT [29]: The HGT model is a learning model proposed for supervised large-scale heterogeneous networks. It uses HG sampling algorithm to compress the size of network. The sampling algorithm can reserve the same number of heterogeneous neighbors and edges for the target node. In order to compare with other unsupervised embedding models, the original supervised loss function in the HGT model is replaced with the loss function consistent with LHGI model. That is, a loss function of maximizing mutual information is used to optimize the parameters in HGT model.

### 4.2. Data Sets

In the baseline models, the random walk process in the Metapath2vec model and Hin2vec model requires higher time consumption and higher storage capacity. The HeGAN model and HDGI model need to learn the embedding of nodes in the initial heterogeneous network. When there are too many nodes in the network, the training cost of these two models will increase exponentially, which will lead to memory overflow and terminate the training process. Therefore, these models cannot be well applied to the learning tasks of heterogeneous networks with millions of nodes. Therefore, three relatively moderate data sets are selected to build heterogeneous networks, so that all models can be trained on these heterogeneous networks. By comparing the classification performance of the node vectors generated by each model on three datasets, the effect of embedding learning of the models is evaluated. The information of the three datasets is as follows:(1)IMDB: IMDB is a movie dataset, which contains three types of nodes, including 4278 movies, 2081 directors and 5257 actors. Among them, movie nodes are divided into three categories: action movies, comedy movies and drama movies.(2)DBLP: DBLP is a citation network dataset, which contains 4 types of nodes, including 4057 authors, 14,376 papers, 20 conferences, and 8920 terms. Among them, author nodes are divided into four categories: database, data mining, information retrieval and machine learning.(3)Yelp: Yelp is a commercial data set, which contains 5 types of nodes, including 2614 business, 1286 users, 2 service types, 2 reservation, and 7 stars levels. Among them, business nodes are divided into three types: Mexican flavor, hamburger type, and food bar.

At the same time, two large-scale heterogeneous networks with millions of nodes are used to discuss the capability of the LHGI model in learning tasks under unsupervised large-scale heterogeneous network.

(4)AMiner: AMiner is a large-scale heterogeneous citation network, which contains 4,888,069 nodes in total. These nodes are specifically divided into 134 venues, 1,693,531 authors and 3,194,405 papers. The author nodes are divided into eight types in AMiner.(5)ogbn-mag: ogbn-mag (Open Graph Benchmark Node Property Prediction-Microsoft Academic Graph) is a heterogeneous graph composed of a subset of the Microsoft Academic Graph. It contains four types of entities, including 736,389 papers, 1,134,649 authors, 8740 institutions, and 59,965 fields of study. The paper nodes are divided into 349 types according to the venue (conference or journal) of each paper.

Table 1 shows the basic information of the five data sets. The last column “Labeled nodes” in Table 1 gives the labeled nodes in each data set, which will be used to test the performance of node embeddings.

### 4.3. Experimental Results

One-hot encoding is used to encode the initial features of nodes. The upper limit of the number of neighbors of each node in the sampling process is set to 20, and the sampling iteration number k is set to 2. Two-layer GCN is used for graph encoding, and four heads of attention are used to fuse the vectors from different metapaths. The value of a batch size is set according to the size of graphics card memory. All nodes in the heterogeneous network participate in the network training and learning process to obtain the embedding vector of each node. In this process, all nodes are regarded as unlabeled, that is, the whole training and learning process is unsupervised.

As shown in Table 2, the performance of different graph convolution strategies is tested first through classification (Micro-F1), clustering (NMI) and the running time of each epoch (second), which will help to determine the best encoder used in LHGI. Three graph convolution strategies of the classical graph convolution [31] encoder (GCNConv), Chebyshev spectral GCN [32] encoder (ChebConv) and GraphSAGE [28] encoder (SAGEConv) are tested. The experimental results show that GCNConv achieves the best comprehensive performance. Although GCNConv runs a little longer in each epoch than SAGEConv, it achieves the best classification and clustering performance on two data sets. Therefore, GCNConv is finally selected as the encoder in the LHGI model.

In order to verify the performance of node embeddings learned from different models, the KNN algorithm is used to classify the labeled nodes in each dataset. K is set to 5 and the number of cycles is 100. Twenty percent of the nodes are used as the training set, and the remaining eighty percent are taken as the verification set. The classification results are evaluated by Macro-F1(F1_mac) and Micro-F1(F1_mic).

Table 3 shows the classification performance of each model on three datasets. It can be seen that the node vectors learned from the LHGI model proposed in this paper obtained the best classification performance on three datasets. It shows that the LHGI model has the best ability to extract features in the network, and the node vectors learned by LHGI can provide more effective information for downstream classification tasks. At the same time, the HGT model based on local structure sampling achieved a classification effect second only to the LHGI model. The HGT model is also a learning model proposed for large heterogeneous networks. It can be seen that, similarly to the LHGI model proposed in this paper, the HGT model also performs well on medium-scale heterogeneous networks.

In order to verify whether the node vectors learned by LHGI model can achieve similar classification performance on other classifiers, the experimental results of the SVM and MLP classifiers are given. Figure 5 shows the classification performance under different classifiers. The classification results are evaluated by Micro-F1 (F1_mic). The results under the KNN classifier are also given in Figure 5 to make a clearer comparison. Experimental results show that the classification performance under SVM and MLP is better than that under KNN. It indicates that the node vectors learned by the LHGI model can achieve good classification performance under different classifiers, which further confirms the ability of the LHGI model to extract network features.

Table 4 shows the clustering performance of each model on three datasets. The K-means algorithm is used for clustering test. The indexes of NMI and ARI are used to evaluate the clustering performance. The number of categories in the cluster is equal to the number of original labels in the dataset. The experimental results show that the LHGI model achieves the best clustering performance on three data sets compared with the baseline models. The clustering results under the LHGI model are more consistent with the data label distribution of the original network.

In the LHGI model, the Discriminator Module adopts the idea of contrastive learning. By destroying the initial characteristic matrix of nodes, LHGI generates negative sample nodes. The lower bound of the mutual information loss function is approximated by the NCE loss between the normal/negative node vectors and the global graph vector with a contrastive learning process, which solves the problem of how to train the model with unsupervised information. At the same time, the metapath guided graph sampling technology proposed in LHGI model can aggregate the semantic information in the metapaths. This enables LHGI model more effective in preserving low-order and high-order semantic information in the network, so that it can learn node vectors with better classification performance.

Among all the baseline models, only the HGT model is proposed for the embedding learning of large-scale heterogeneous network. In order to compare the embedding effects of the HGT model and LHGI model on large-scale heterogeneous networks, the classification performance of the vectors generated by the two models was analyzed. The Softmax function of MLP was used to achieve the classification results. Figure 6 shows the classification results of the two models on AMiner and ogbn-mag. From the experimental results, we can see that both models can train and learn large-scale heterogeneous networks. However, compared with the HGT model, the LHGI model achieved better classification performance on both large-scale data sets.

The HGT model directly samples the local structure from the initial heterogeneous network, which makes the learning process depend only on the sampled local substructure. The LHGI model adds the guidance of the metapath relationship in the sampling process, which can help preserve the higher-order and deeper semantic relationships in the network as much as possible. Therefore, compared with HGT model, LHGI can reduce the loss of semantic information in the network as much as possible. This enables the node vectors learned by the LHGI model to contain richer semantic information, so as to better perform the downstream mining tasks.

## 5. Conclusions

In order to solve the problem of embedding learning in unsupervised large-scale heterogeneous networks, this paper proposes an embedding learning model, LHGI, based on metapath guided graph sampling. The LHGI model consists of the Meta Path Generating Module, Meta Path Guided Graph Sampling Module, Sampling Graph Aggregation Encoding Module and Discriminator Module, which respectively perform the tasks of automatically generating metapaths, graph sampling guided by metapaths, node (graph) encoding and maximizing the mutual information. To solve the problem of embedding learning in large-scale heterogeneous networks, the LHGI model adopts the strategy of graph sampling under the guidance of metapaths to compress the size of the network. Metapaths themselves are structural sequences in the network, which contain rich semantic information. The graph sampling process based on metapaths can realize the iterative fusion of the semantic information in metapaths. This process enables the LHGI model to capture the local and global semantic information in the heterogeneous network as much as possible, thus reducing the loss of information as much as possible. In order to solve the problem of embedding learning without supervised information, the LHGI model adopts the idea of contrastive learning. The LHGI model generates some negative sample nodes by scrambling the characteristic matrix. The loss function of the LHGI model is set by maximizing the mutual information between vectors of normal nodes and the overall graph, and minimizing the mutual information between vectors of negative sample nodes and the graph, so that the model can be trained even without supervised information. Experimental results show that the LHGI model is well qualified for embedding tasks on large-scale heterogeneous networks with millions of nodes. At the same time, the LHGI model is also suitable for learning tasks on medium-scale heterogeneous networks. Even without supervised information, the LHGI model successfully completed the task of network feature extraction. The node vectors learned by the LHGI model perform well in the downstream mining tasks.

In this paper, the classification and clustering tasks are used to verify the performance of the node vectors learned by the models. In fact, in addition to these tasks, the LHGI model can also be applied to other downstream mining tasks, such as the link prediction task, which is used to determine the possibility of connecting edges between two nodes [21]. On the basis of the node vectors generated by the LHGI model, the edge features can be learned by binary operation on the node vectors. Then, the link prediction task can be transformed into a classification task to predict the probability of connecting edges between nodes.

At the same time, a heterogeneous network is actually a special case of hypergraph. In a heterogeneous network, one edge can only connect two nodes, while in a hypergraph, one edge can connect multiple nodes. This makes the embedding learning process in hypergraph more complicated than that in heterogeneous network [33,34]. In a hypergraph, it needs to aggregate the features of nodes connected by one hyperedge to generate the hyperedge character. Then, the hyperedge characters are aggregated to learn the embedding of the target node. The idea of aggregating neighboring nodes (edges) in the hypergraph is the same as that used in the LHGI model, which uses GCN to aggregate the characteristics of neighboring nodes. In future work, it would be interesting to explore the application of LHGI model in hypergraph embedding learning. At the same time, we will further discuss the application of the LHGI model in the downstream mining tasks of hypergraph, such as hypergraph-based classification or link prediction tasks.

## Figures and Tables

**Figure 1 entropy-25-00297-f001:**
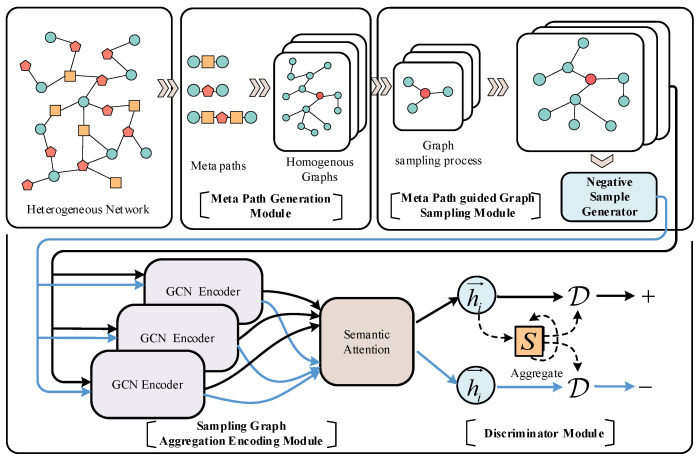
Framework of the LHGI model.

**Figure 2 entropy-25-00297-f002:**
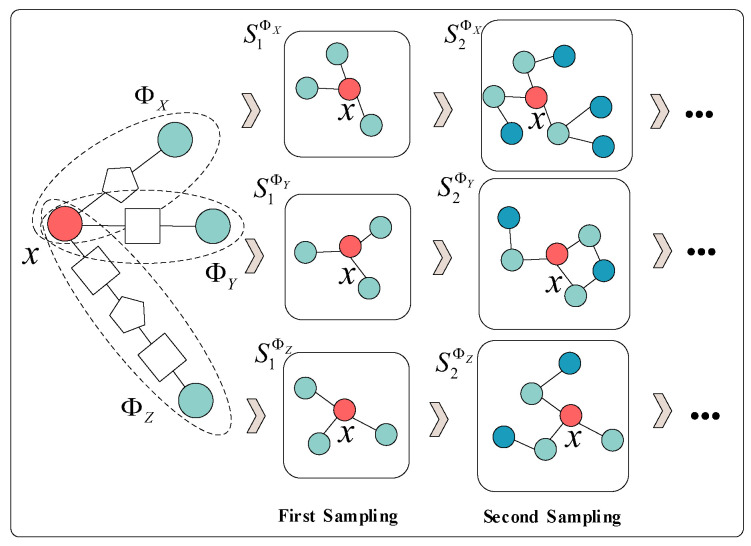
Schematic diagram of the graph Sampling process guided by metapaths.

**Figure 3 entropy-25-00297-f003:**
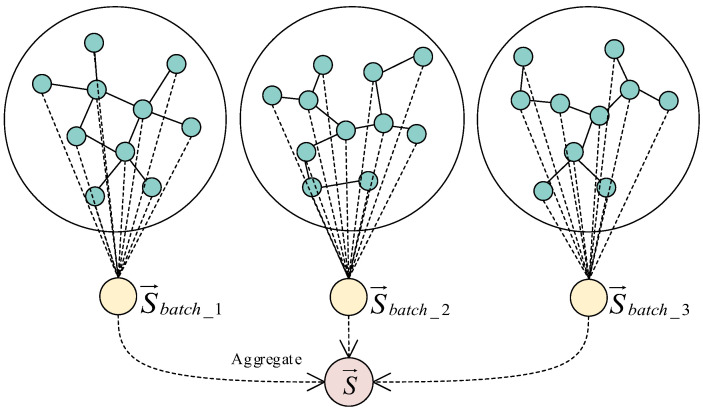
Schematic diagram of generating the global map vector.

**Figure 4 entropy-25-00297-f004:**
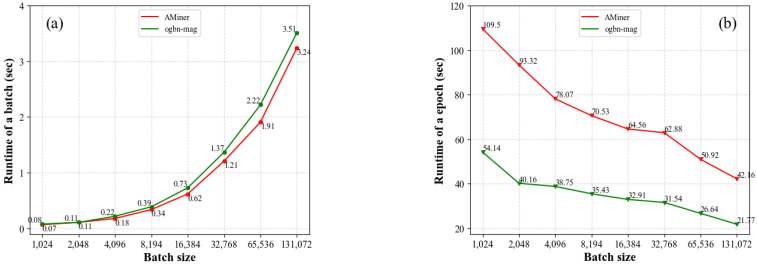
Relationship between batch size and batch runtime (**a**)/epoch runtime (**b**).

**Figure 5 entropy-25-00297-f005:**
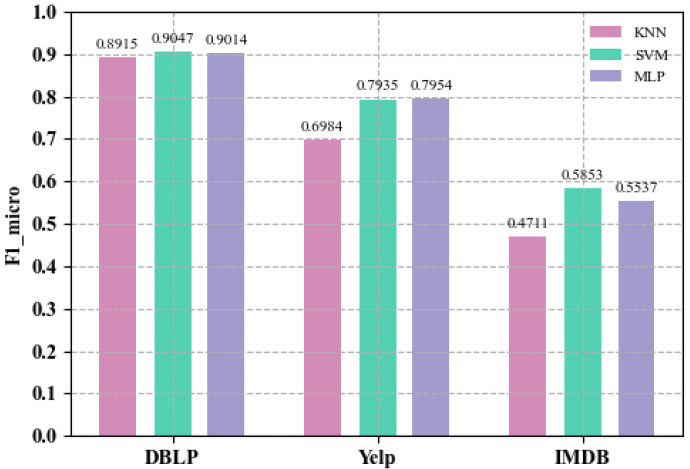
Classification performance of node vectors learned by LHGI under different classifiers.

**Figure 6 entropy-25-00297-f006:**
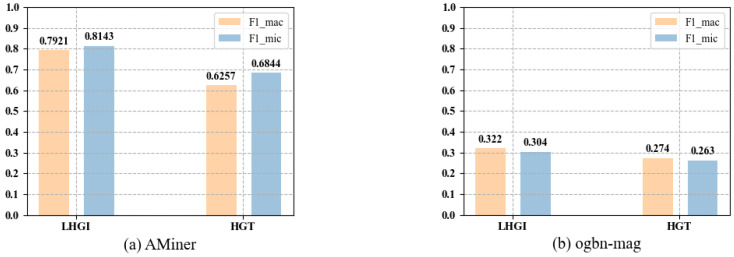
Classification performance on the large-scale heterogeneous network of AMiner (**a**)/ogbn-mag (**b**).

**Table 1 entropy-25-00297-t001:** The basic information of the data sets.

Dataset	Node Type	Nodes	Edge Type	Edges	Labeled Nodes
IMDB	Movie(M) Director(D) Actor (A)	4278 2081 5257	M-D M-A D-M A-M	4278 12,828 4278 12,828	Movie (3)
DBLP	Author (A) Conference (C) Paper (P) Term (T)	4057 20 14,376 8920	P-A P-C P-T	19,645 14,376 114,625	Author (4)
Yelp	Business (B) Reservation (R) Service (S) Stars Level (SL) User (U)	2614 2 2 9 1286	B-SL B-R B-S B-U	2614 2614 2614 30,839	Business (3)
AMiner	Author (A) Paper (P) Venue (V)	1,693,531 3,194,405 134	A-P P-V	9,323,605 3,194,405	Author (8)
ogbn-mag	Author (A) Paper (P) Institutions (I) fields of study (F)	1,134,649 736,389 8740 59,965	P-P P-F A-P A-I	5,416,271 7,505,078 7,145,660 1,043,998	Paper (349)

**Table 2 entropy-25-00297-t002:** Comparative analysis of three graph convolution strategies.

HLGI	DBLP			Yelp			IMDB		
	F1_mic	NMI	Time	F1_mic	NMI	Time	F1_mic	NMI	Time
GCNConv	0.8915	0.6188	5.740	0.6984	0.3510	6.722	0.4711	0.0427	5.401
ChebConv	0.8766	0.5644	6.768	0.6761	0.3431	7.876	0.4928	0.0755	6.207
SAGEConv	0.6307	0.5918	4.457	0.6689	0.3480	5.229	0.4979	0.0299	4.188

**Table 3 entropy-25-00297-t003:** Classification performance of each model.

MODEL	DBLP		Yelp		IMDB		ALL_DATA	
	F1_mac	F1_mic	F1_mac	F1_mic	F1_mac	F1_mic	mac_avg	mic_avg
M2V	0.6985	0.6874	0.4534	0.5171	0.3933	0.4051	0.515	0.5365
Hin2vec	0.605	0.594	0.4011	0.3541	0.325	0.3261	0.4437	0.4247
HeGAN	0.7544	0.7702	0.4578	0.5264	0.4057	0.4177	0.5393	0.5714
HDGI	0.7153	0.7259	0.4096	0.4429	0.4445	0.4466	0.5231	0.5384
HGT	0.6403	0.6885	0.5917	0.6775	0.386	0.3882	0.5393	0.5847
LHGI	0.884	0.8915	0.625	0.6984	0.4650	0.4711	0.6600	0.6850

**Table 4 entropy-25-00297-t004:** Clustering performance of each model.

MODEL	DBLP		YELP		IMDB		ALL_DATA	
	NMI	ARI	NMI	ARI	NMI	ARI	NMI _avg	ARI _avg
M2V	0.4577	0.4806	0.1102	0.1443	0.0115	0.0151	0.1931	0.2133
Hin2vec	0.442	0.4699	0.2324	0.24	0.0102	0.0105	0.2282	0.2401
HeGAN	0.5546	0.572	0.2544	0.2608	0.0366	0.0376	0.2818	0.2901
HDGI	0.6076	0.6076	0.2334	0.2011	0.0187	0.037	0.2865	0.2882
HGT	0.4603	0.4712	0.2018	0.2146	0.0124	0.0165	0.2248	0.2341
HLGI	0.6188	0.6203	0.3510	0.4018	0.0427	0.0241	0.3375	0.3487

## Data Availability

Publicly available datasets were analyzed in this study. This data can be found here: [https://pytorch-geometric.readthedocs.io/en/latest/modules/datasets.html], accessed on 9 January 2023.

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
