# Peer review of "Unsupervised Embedding Learning for Large-Scale Heterogeneous Networks Based on Metapath Graph Sampling"

_entropy, 2023, doi:10.3390/e25020297_

Round 1

Reviewer 1 Report

The authors proposed a new method called LHGI for unsupervised embedding learning in large-scale heterogeneous networks by exploiting metapaths. The paper is well written. 

Comments:

1. The paper is missing some important references, e.g., 

a. Heterogeneous Network Representation Learning: A Unified Framework with Survey and Benchmark.

b. Heterogeneous Network Representation Learning.

2. Why did the authors choose the attention-based GCN (1) and (2)? How about other advanced GCNs?

3. Can the authors apply LHGI for link prediction as in "Heterogeneous Network Representation Learning: A Unified Framework with Survey and Benchmark"?

4. The authors claimed that their method can be applied to large-scale heterogeneous networks. Can the authors perform complexity analysis for LHGI and test it on large-scale networks?

5.  How did the authors perform validation for LHGI and other baseline methods? How much training, validation, and test during the experiments?

6. The authors should test LHGI on other classifiers like SVM and MLP? Can obtain similar results?

7. The authors should discuss the future work of LHGI in the conclusion. For example, how to extend LHGI to hypergraphs and apply for hypergraph node classification and hyperlink prediction? Check the references like

a. Hypergraph Learning: Methods and Practices.

b. A Survey on Various Representation Learning of Hypergraph for Unsupervised Feature Selection.

c. A Survey on Hyperlink Prediction. 

Author Response

Reviewer#1, Concern # 1:

Comments:

  1. The paper is missing some important references.

Our response:

Thank you very much for your constructive suggestions.

       These important references have been added in the revised version.

  1. Why did the authors choose the attention-based GCN (1) and (2)? How about other advanced GCNs?

Our response:

Thank you very much for your constructive suggestions.

GCN can capture the local topology characteristics of nodes in the network without prior information, so GCN is used to learn the embedding vector of nodes in our work. But for a given node, it may appear in different meta paths, which means that one node will obtain different embeddings by GCN. Thus, the semantic attention mechanism is used to measure the weight of semantic information provided by different meta paths. According to the weight of the meta paths, the node vectors generated by different meta paths are aggregated to obtain the final embedding of the node. Therefore, in our work, GCN is first used to generate node vectors, and then attention mechanism is taken to fuse the node vectors. In the revised version, we further elaborated the process in detail.

  1. Can the authors apply LHGI for link prediction as in "Heterogeneous Network Representation Learning: A Unified Framework with Survey and Benchmark"?

Our response:

Thank you very much for your constructive suggestions.

In this paper, the classification and clustering tasks are used to verify the performance of the node vectors learned by different models. Experimental results show that the node vectors learned by LHGI model performs best in classification and clustering tasks. Link prediction is also an interesting downstream mining task to test the node embeddings. In the future, we will continue to apply LHGI for link prediction as in "Heterogeneous Network Representation Learning: A Unified Framework with Survey and Benchmark".

Thank you for the valuable suggestion again.

  1. The authors claimed that their method can be applied to large-scale heterogeneous networks. Can the authors perform complexity analysis for LHGI and test it on large-scale networks?

Our response:

Thank you very much for your constructive suggestions.

In section 3.5 of the revised version, we added the analysis of the spatial complexity and time complexity of LHGI model.

The spatial complexity of LHGI model is mainly determined by the Meta Path Generation Module, Meta Path guided Graph Sampling Module, and Sampling Graph Aggregation Encoding Module. The number of meta-paths obtained in the Meta Path Generation Module determines the number of GCN encoders required by the LHGI model. In the Meta Path guided Graph Sampling Module, the number of target nodes processed by each batch, the number of neighbor nodes sampled by each batch, and the number of sampling batches are the main factors affecting the input scale of the LHGI model. In the Sampling Graph Aggregation Encoding Module, LHGI model uses a two-layer GCN encoder to avoid the over-smoothing problem in the learning process. Suppose the number of meta paths is a, the number of target nodes processed in each batch is b, the number of neighbors sampled in each batch is c, and the number of sampling iterations is d (which is equal to the number of layers of GCN encoder by default), then the spatial complexity of LHGI model is O(a(bc)d).

The time complexity of LHGI model is determined by the running time of each batch and the number of batches. The number of batches is related to the number of target nodes processed under each batch. The more target nodes processed in each batch, the less batches required, and vice versa. Therefore, the time complexity of LHGI model can be expressed as O(b-1t) , where b is the number of target nodes processed by each batch, and t is the running time for each batch. Figure 4 shows the relationship between batch size and batch (epoch) runtime. Experimental results show that for large-scale heterogeneous networks, LHGI model can easily train the network under different batch sizes. In practical applications, the appropriate batch size can be selected according to the hardware environment, which enables LHGI model more universal.

  1. How did the authors perform validation for LHGI and other baseline methods? How much training, validation, and test during the experiments?

Our response:

Thank you very much for your constructive suggestions.

All nodes in the heterogeneous network participate in the network training and learning process to obtain the embedding vector of each node. In this process, all nodes are regarded as unlabeled, that is, the whole training and learning process is unsupervised.

In order to verify the performance of node embeddings learned from different models, KNN algorithm is used to classify the labeled nodes in each dataset. k is set to 5 and the number of cycles is 100. 20% nodes are used as training set, and the remaining 80% are taken as the verification set. The classification results were evaluated using Macro-F1 and Micro-F1.

In the revised version, we further elaborated the process in detail.

  1. The authors should test LHGI on other classifiers like SVM and MLP? Can obtain similar results?

Our response:

Thank you very much for your constructive suggestions.

In section 4.3 of the revised version, we added Figure 5 to show the classification results of the node vectors learned from LHGI model on the SVM and MLP classifiers. The results under KNN classifier are also given in Figure 5 to make a clearer comparison. Experimental results show that the classification performance under SVM and MLP is better than that under KNN. It indicates that the node vectors learned by LHGI model can achieve good classification performance under different classifiers, which further confirms the ability of LHGI model to extract network features.

  1. The authors should discuss the future work of LHGI in the conclusion. For example, how to extend LHGI to hypergraphs and apply for hypergraph node classification and hyperlink prediction? Check the references like:
  2. Gao Y, Zhang Z Z, Lin H J, Zhao X B, Du S Y, etc. Hypergraph Learning: Methods and Practices. IEEE Transactions on Pattern Analysis and Machine Intelligence, 44(5): 2548-2566.
  3. A Survey on Various Representation Learning of Hypergraph for Unsupervised Feature Selection.
  4. Chen C., Liu Y. Y. A Survey on Hyperlink Prediction. arXiv:2207.02911v1, 2022.

Our response:

Thank you very much for your constructive suggestions.

Heterogeneous network is actually a special case of hypergraph. In a heterogeneous network, one edge can only connect two nodes. While in a hypergraph, one edge can connect multiple nodes. This makes the embedding learning process in hypergraph more complicated than that in heterogeneous network. In a hypergraph, it needs to aggregate the features of nodes connected by one hyperedge to generate the hyperedge character. Then, the hyperedge characters are aggregated to learn the embedding of target node. The idea of aggregating neighboring nodes (edges) in the hypergraph is the same as that used in the LHGI model, which uses GCN to aggregate the characteristics of neighboring nodes. In future work, it is interesting to explore the application of LHGI model in hypergraph embedding learning. At the same time, we will further discuss the application of LHGI model in the downstream mining tasks of hypergraph, such as hypergraph-based classification or link prediction tasks.

We have added these discussions in the Conclusion section of the revised version.

Thanks a lot again.

Reviewer 2 Report

This manuscript presents a novel technique for embedding large heterogeneous graphs. The novelties include a construction of homogenous graphs, a sampling technique, and an unsupervised mechanism based on mutual information. All the methods introduced are incremental. The approach is interesting and effective, as demonstrated by the results, but there are some issues to address in order to improve the paper:

1) In the meta-path generation module, homogeneous subgraphs are constructed out from the original heterogeneous path by applying four rules. These are described only in word, but not in pseudocode or (better) in mathematical notation.

2) The graph sampling procedure based on meta-graphs (and the example in Figure 2) is confusing. Is this sampling with replacement? How are the samplings in Figures 2 obtained? Explain with finer detail and more clarity the sampling process

3) Similar to the above, the negative sampling model is only explained superficially.

4) What is the approximate computational complexity in the procedure? The use of several GCNs points at a quite intensive computation algorithm

5) Despite what the title suggests, only in one of the experimental datasets (AMiner) the case is actually a large scale network with millions of nodes. 

6) The numerical results are quite limited, only two performance metrics are used. Can this be improved to include other performance metrics for comparison?

Author Response

Reviewer#2, Concern # 1:

Comments:

This manuscript presents a novel technique for embedding large heterogeneous graphs. The novelties include a construction of homogenous graphs, a sampling technique, and an unsupervised mechanism based on mutual information. All the methods introduced are incremental. The approach is interesting and effective, as demonstrated by the results, but there are some issues to address in order to improve the paper:

  1. In the meta-path generation module, homogeneous subgraphs are constructed out from the original heterogeneous path by applying four rules. These are described only in word, but not in pseudocode or (better) in mathematical notation.

Our response:

Thank you very much for your constructive suggestions.

In section 3.2.1 of the revised version, we added the mathematical expressions about the meta-path screening principles.

  1. The graph sampling procedure based on meta-graphs (and the example in Figure 2) is confusing. Is this sampling with replacement? How are the samplings in Figures 2 obtained? Explain with finer detail and more clarity the sampling process.

Our response:

Thank you very much for your constructive suggestions.

We’re sorry not to clearly explain the process of graph sampling under the guidance of meta path in the manuscript. In the revised version, we added a more detailed discussion of Figure 2. We hope that these discussions can be more helpful to understand the process of graph sampling.

By taking the target node x as an example, Figure 2 shows how to sample graph under the three meta paths where x acts as the head or tail node. The first sampling will obtain the first-order neighbor of x. These neighbors are not the real first-order neighbors of x in the initial heterogeneous network, but the neighbor nodes that have the head-to-tail correspondence with x in the meta-path. Next, taking the neighbor nodes obtained in the first sampling as seed nodes, the neighbor nodes of these seed nodes under the meta-path are sampled to generate the second sampling set. The second sampling will obtain the second-order neighbors of x in the homogeneous graph. Similarly, these neighbors are not the initial second-order neighbors, but the second-order neighbors in the sense of meta-path. Because the sampling graph is obtained by sampling the meta path for many times, the sampling process can approach the deeper semantic information contained in the initial heterogeneous network as much as possible. This enables LHGI model more effective in preserving low-order and high-order semantic information in the network, so that it can learn node vectors with better performance in the downstream mining tasks.

  1. Similar to the above, the negative sampling model is only explained superficially.

Our response:

Thank you very much for your constructive suggestions.

In the revised version, we added a more detailed discussion on the negative sampling module.

In LHGI model, a Negative Sample Generator is used to generate negative sample nodes by randomly disrupting the rows in the initial node characteristic matrix X, which will disrupt the order of node indexes and destroy the connection between nodes. Then, the same embedding learning process as the normal sample nodes is performed to learn the embedding vectors of the negative sample nodes. Because the initial characteristics of the negative sample nodes and their adjacent relationship with other nodes are randomly disturbed, the negative sample node vector is not learned on the real network space features. This makes the vector of the negative sample node significantly different from that of the normal node, so the LHGI model can solve the problem of unsupervised training through the idea of comparative learning.

  1. What is the approximate computational complexity in the procedure? The use of several GCNs points at a quite intensive computation algorithm.

Our response:

Thank you very much for your constructive suggestions.

In section 3.5 of the revised version, we added the analysis of the spatial complexity and time complexity of LHGI model.

The spatial complexity of LHGI model is mainly determined by the Meta Path Generation Module, Meta Path guided Graph Sampling Module, and Sampling Graph Aggregation Encoding Module. The number of meta-paths obtained in the Meta Path Generation Module determines the number of GCN encoders required by the LHGI model. In the Meta Path guided Graph Sampling Module, the number of target nodes processed by each batch, the number of neighbor nodes sampled by each batch, and the number of sampling batches are the main factors affecting the input scale of the LHGI model. In the Sampling Graph Aggregation Encoding Module, LHGI model uses a two-layer GCN encoder to avoid the over-smoothing problem in the learning process. Suppose the number of meta paths is a, the number of target nodes processed in each batch is b, the number of neighbors sampled in each batch is c, and the number of sampling iterations is d (which is equal to the number of layers of GCN encoder by default), then the spatial complexity of LHGI model is O(a(bc)d).

The time complexity of LHGI model is determined by the running time of each batch and the number of batches. The number of batches is related to the number of target nodes processed under each batch. The more target nodes processed in each batch, the less batches required, and vice versa. Therefore, the time complexity of LHGI model can be expressed as O(b-1t) , where b is the number of target nodes processed by each batch, and t is the running time for each batch. Figure 4 shows the relationship between batch size and batch (epoch) runtime. Experimental results show that for large-scale heterogeneous networks, LHGI model can easily train the network under different batch sizes. In practical applications, the appropriate batch size can be selected according to the hardware environment, which enables LHGI model more universal.

  1. Despite what the title suggests, only in one of the experimental datasets (AMiner) the case is actually a large scale network with millions of nodes.

Our response:

Thank you very much for your constructive suggestions.

In the revised version, we added another large-scale heterogeneous network, ogbn-mag. It is a heterogeneous graph composed of four types of entities, including 736,389 papers, 1,134,649 authors, 8,740 institutions, and 59,965 fields of study. The LHGI model and baseline model are trained to learn the embedding vector of nodes in ogbn-mag. And the classification performance of the node vectors learned by each model is verified. The experimental results show that LHGI model achieves better classification performance on ogbn-mag than the baseline model. It indicates that LHGI model successfully extracts the features in the new heterogeneous network, and obtains the node vector rich in semantic information. Figure 6 in the revised version gives the classification results.

And in section 3.2.5 of the revised manuscript, we discussed the spatial and time complexity of LHGI model. By taking the two large-scale heterogeneous networks of AMiner and ogbn-mag as example, Figure 4 gives the relationship between batch size and batch (epoch) runtime. Experimental results show that for large-scale heterogeneous networks, LHGI model can easily train the network under different batch sizes.

  1. The numerical results are quite limited, only two performance metrics are used. Can this be improved to include other performance metrics for comparison?

Our response:

Thank you very much for your constructive suggestions.

In the revised version, we added a comparative analysis of the performance of the node vectors learned from each model under the clustering task. The experimental results are shown in Table 3 in the revised version. It shows that LHGI model achieves best clustering performance on three data sets compared with the baseline models. The clustering results under LHGI model are better consistent with the data label distribution of the original network, which proves that LHGI model has a better ability to extract features from heterogeneous networks.

Thanks a lot again.

Round 2

Reviewer 1 Report

Thank you for addressing the comments. Further comments:

1. Regrading Comment 2, did the authors consider other advanced GCNs like GraphSAGE or Chebyshev spectral GCN? The authors should try them to see if they can achieve better performances than the classical GCN (Equation (1)). 

2. Regarding Comment 7, it is recommended that the authors can include the mentioned hypergraph references when discussing aggregating node/edge embeddings since it is very commonly used in hypergraph learning.

Author Response

Reviewer#1:

Comments:

  1. Regrading Comment 2, did the authors consider other advanced GCNs like GraphSAGE or Chebyshev spectral GCN? The authors should try them to see if they can achieve better performances than the classical GCN (Equation (1)).

Our response:

Thank you very much for your constructive suggestions.

       In the revised version, we added a comparative experiment on the performance of LHGI model in classification, clustering and running time under different convolution strategies to help select the best encoder for LHGI model. Three graph convolution strategies of classical graph convolutionencoder (GCNConv), Chebyshev spectral GCN encoder (ChebConv) and GraphSAGE encoder (SAGEConv) are tested. The experimental results show that GCNConv has achieved the best comprehensive performance. Although GCNConv runs a little longer in each epoch than SAGEConv, it achieves the best classification and clustering performance on two data sets. Therefore, GCNConv is finally selected as the encode in the LHGI model. The detailed experimental results are shown in Table 2 in the revised version.

  1. Regarding Comment 7, it is recommended that the authors can include the mentioned hypergraph references when discussing aggregating node/edge embeddings since it is very commonly used in hypergraph learning.

Our response:

Thank you very much for your constructive suggestions.

In the conclusion of the revised version, we added these important references.

Thanks a lot again.

Reviewer 2 Report

The authors have addressed and responded clearly and properly to all the previous concerns. I believe the contribution is valuable and the technical content of the paper is good, therefore publication of this work is recommended.

Author Response

Reviewer#2:

Comments:

The authors have addressed and responded clearly and properly to all the previous concerns. I believe the contribution is valuable and the technical content of the paper is good, therefore publication of this work is recommended.

Our response:

Thank you very much for your approval of our work. Thanks a lot again.